Conservation genetics of extremely isolated urban populations of the northern dusky salamander (Desmognathus fuscus) in New York City

Munshi-South Jason 1 2 jason@NYCevolution.org
Zak Yana 1
Pehek Ellen 3
1 Department of Natural Sciences, Baruch College, City University of New York (CUNY) , New York, NY , USA
2 Program in Ecology, Evolutionary Biology & Behavior, The Graduate Center, City University of New York , New York, NY , USA
3 New York City Department of Parks and Recreation, Natural Resources Group, Urban Field Station , Bayside, NY , USA
Crandall Keith
Electronic publication date: 2013 Apr 9
Publication date: 2013
Volume: 1
Electronic Location ID: e64
Received 2013 Feb 15; Accepted 2013 Mar 18
Copyright: © 2013 Munshi-South et al.
Copyright year: 2013
Copyright holder: Munshi-South et al.
License: This is an open access article distributed under the terms of the Creative Commons Attribution License, which permits unrestricted use, distribution, and reproduction in any medium, provided the original author and source are credited.
License URL: https://creativecommons.org/licenses/by/3.0/

Keywords: Genetic variation, Stream salamander, Plethodontidae, Urban ecology, Microsatellite, Genetic structure, Urban evolutionary biology

Funding: National Science Foundation DEB-0817259 This research was supported by National Science Foundation grant DEB-0817259 to Jason Munshi-South. The funders had no role in study design, data collection and analysis, decision to publish, or preparation of the manuscript.

==============================
Urbanization is a major cause of amphibian decline. Stream-dwelling plethodontid salamanders are particularly susceptible to urbanization due to declining water quality and hydrological changes, but few studies have examined these taxa in cities. The northern dusky salamander (Desmognathus fuscus) was once common in the New York City metropolitan area, but has substantially declined throughout the region in recent decades. We used five tetranucleotide microsatellite loci to examine population differentiation, genetic variation, and bottlenecks among five remnant urban populations of dusky salamanders in NYC. These genetic measures provide information on isolation, prevalence of inbreeding, long-term prospects for population persistence, and potential for evolutionary responses to future environmental change. All populations were genetically differentiated from each other, and the most isolated populations in Manhattan have maintained very little genetic variation (i.e. <20% heterozygosity). A majority of the populations also exhibited evidence of genetic bottlenecks. These findings contrast with published estimates of high genetic variation within and lack of structure between populations of other desmognathine salamanders sampled over similar or larger spatial scales. Declines in genetic variation likely resulted from population extirpations and the degradation of stream and terrestrial paths for dispersal in NYC. Loss of genetic variability in populations isolated by human development may be an underappreciated cause and/or consequence of the decline of this species in urbanized areas of the northeast USA.

Introduction

Urbanization has emerged as a substantial cause of the decline of amphibian species (Gibbs, Whiteleather & Schueler, 2005; Hamer & McDonnell, 2008). Increasing human population density produces habitat loss and fragmentation that are implicated in the decline of all but a few generalist species that can survive in the urban matrix. Roads alone result in increased mortality and habitat fragmentation (Balkenhol & Waits, 2009), and are potentially potent barriers to gene flow between some amphibian populations (Emel & Storfer, 2012). Over 30 studies have been conducted on urban amphibians in North America, but research has been biased toward anuran sampling using call surveys (Scheffers & Paszkowski, 2011). Lungless salamanders (Family: Plethodontidae) are particularly understudied, despite their potential as bioindicators of habitat quality (Welsh & Droege, 2001). The few plethodontid studies indicate that species richness and abundance of individual stream-dwelling species decline after urbanization (Hamer & McDonnell, 2008; Price et al., 2011). Experimental data suggest that increased impermeable surfaces in urban watersheds result in larval salamanders being flushed out of streams at a higher rate due to increased water velocity (Barrett et al., 2010). Furthermore, riparian buffer zones provide little protection in urbanized watersheds (Willson & Dorcas, 2003). Nearly half of New York City’s (NYC) native salamander species have been lost over the last century. Stream dwelling species have fared slightly better than pond breeding taxa, perhaps due to the option of dispersing through stream networks rather than overland through degraded urban matrix (Pehek, 2007).

The northern dusky salamander, Desmognathus fuscus (Rafinesque, 1820), persists in small, isolated streams and spring-fed seeps that have escaped development in the NYC metropolitan area. Populations in Manhattan persist in narrow, linear parklands (i.e. less than 200 m wide) on rocky hillsides surrounded by roads and urban development, whereas Staten Island and suburban New Jersey populations inhabit streams and swamps in larger protected areas of secondary forest that may still be connected through stream networks. This species was common in NYC as late as the 1950 (Kieran, 1982), but has since declined regionally due to habitat destruction and declining water quality. Population losses have also been reported from large, contiguous protected areas in the northeastern USA (Bank et al., 2006), and population densities of southern dusky salamanders (D. auriculatus) are inversely correlated with the degree of urbanization (Orser & Shure, 1972). Population stability may be realized when salamanders disperse through multiple pathways in stream networks (Campbell Grant et al., 2010), but both streams (Walsh et al., 2005) and overland corridors (Munshi-South, 2012) are typically degraded in urban environments. Other small vertebrates with limited dispersal ability also exhibit population genetic patterns consistent with loss of population connectivity. Terrestrial plethodontids (Marsh et al., 2007; Noël & Lapointe, 2010), small mammals (Munshi-South & Kharchenko, 2010), birds, and lizards (Delaney, Riley & Fisher, 2010) all exhibit substantial genetic differentiation and isolation among isolated urban populations.

In this study, we use five tetranucleotide microsatellite loci to examine genetic diversity and differentiation among D. fuscus populations inhabiting isolated streams in NYC. We sampled all known locations in NYC and contiguous suburban counties to provide context on the genetic relationships between NYC populations and surrounding regions. D. fuscus populations in NYC are well-known to the local human communities surrounding their stream habitats, and are a species of interest for conservation efforts in northern Manhattan and the Staten Island Greenbelt watershed. While these salamanders have persisted for decades in the same highly urbanized locations (Gans, 1945), loss of genetic variation due to inbreeding and isolation are unknown. Due to its regional population decline, degradation of stream networks, and sensitivity to urban water quality, we predicted that D. fuscus exhibits even greater genetic impacts from urbanization than other small vertebrates with limited mobility. Under this scenario, remnant D. fuscus populations may require conservation efforts to restore connectivity between populations or translocations to counteract inbreeding depression. This study is the first to our knowledge to investigate the conservation genetics of urban stream salamanders, and the results will be applied to future amphibian conservation decisions concerning amphibian reintroductions, watershed management, and landscape planning for biodiversity in NYC.

Methods

Sampling was conducted in 2010–2011 at five sites known to harbor dusky salamanders (Fig. 1): two seepage areas in Highbridge Park, Manhattan, NYC; streams in Corson’s Brook Woods and Reed’s Basket Willow Swamp Park in Staten Island, NYC; and a stream in the Watchung Reservation, Union Co., NJ, approximately 48 km west of NYC. These sites were sampled because they were the only sites known to contain D. fuscus in NYC or adjacent suburban counties, and the NYC sites are managed by the NYC Department of Parks & Recreation as “Forever Wild” conservation areas. The Highbridge Park sites were rediscovered by one of the authors (EP) in 2005 based on a note in the herpetological literature (Gans, 1945). These two seeps are isolated from each other by the Washington and Hamilton Bridges (erected in 1888 and 1963, respectively) that cumulatively carry fourteen lanes of automobile traffic. The Staten Island sites are part of the Greenbelt, a contiguous series of protected areas totaling 1,100 ha composed of swamps and secondary forests. The Watchung Reservation in NJ is a 790 ha protected area composed of primary forest and recreational areas in a valley of the Watchung mountains, and is surrounded by low- to medium-density suburban housing. We also searched for but were unable to locate D. fuscus at other locations where they have been reported in recent decades, including the Lost Brook Preserve in the Palisades, NJ, and two sites in Westchester Co., NY: Hilltop Hanover Farms and Ward Pound Ridge Reservation. We initially also failed to sample D. fuscus from the Great Swamp at the Greenbelt Nature Center, Staten Island, NY, but did find a few individuals after the sampling and lab work for this study were concluded. These samples and another site recently discovered on Staten Island (Bloodroot Valley) will be included in a future landscape genomics study of urban salamanders (see Discussion).

Figure 1 Map of study sites.

Map of study sites in relation to urbanization in the NYC metropolitan area. Areas colored in shades of red and purple denote landscape areas with increasingly greater percentages of impervious surfaces as measured by the 2006 National Landcover Database (Homer et al., 2004).

We captured salamanders by hand or dipnet after turning over cover objects such as logs, rocks, bottles, and metal/plastic sheeting in or near streams. Tail tips were clipped and stored in 80% ethanol until DNA extraction using the standard protocol for the Qiagen DNEasy tissue kit. All animal handling protocols were approved by the Natural Resources Group of the NYC Department of Parks and Recreation, and followed the recommendations of the Declining Amphibian Task Force’s “Fieldwork Code of Practice” (http://www.fws.gov/ventura/species_information/protocols_guidelines/docs/DAFTA.pdf) and the NY State Department of Environmental Conservation’s “Bio-safety Protocols for Reptile and Amphibian Sampling”. Five previously described microsatellite loci were PCR-amplified in 15 µl volumes using Promega PCR master mix and published thermal cycling profiles. PCR included one primer with a CAG or M13R tail and an associated probe with fluorescent WellRED D2, D3, or D4 dye. The total reaction included 7.5 µl master mix, 4.4 µl water, 0.8 µl of the 10 µM untailed primer, and 0.4 µl each of the 10 µM tailed primer and 10 µM fluorescent probe. The amplified loci included Dau3, Dau11, and Dau12 from Croshaw & Glenn (2003), Doc03 from Adams, Jones & Arnold (2005), and ENS6 from Devitt et al. (2009). PCR fragments were separated and sized on a Beckman Coulter CEQ8000 sequencer. Alleles were scored using automatic binning procedures followed by visual inspection in the Beckman fragment analysis software. The genotypes and spatial coordinates for all study sites are available on the Dryad digital repository (DOI 10.5061/dryad.q1nc0).

Each locus was tested for deviations from Hardy–Weinberg (HWE) and linkage equilibrium over the total sample of 141 genotypes and within each of the five sampling sites using GENEPOP 4.0 (Rousset, 2008). We also used MICRO-CHECKER to analyze genotypes within each population for homozygote excess due to null alleles, allelic dropout, or errors in allele calling due to stuttering (van Oosterhout et al., 2004). To characterize genetic diversity, we calculated the numbers of alleles, effective alleles, and private alleles at each site, and the observed and expected heterozygosity at each site and for each locus across the entire sample, using GenAlex 6.4 (Peakall & Smouse, 2005). We tested for genetic bottlenecks in each population using the authors’ recommended settings for microsatellites (TPM; 95% single-step mutations) in BOTTLENECK 1.2 (Piry, Luikart & Cornuet, 1999).

To examine population differentiation, we calculated pairwise FST between all site pairs using 1,000 random permutations in GenAlex to assess significance. We then used the evolutionary clustering method implemented in STRUCTURE 2.3 to place individual genotypes in clusters that minimized deviations from Hardy–Weinberg and linkage equilibria (Pritchard, Stephens & Donnelly, 2000). We did not use the sampling site as prior information, and allowed for correlated allele frequencies and genetic admixture across populations (Falush, Stephens & Pritchard, 2003). We conducted ten replicate runs for each value of K = 1–10, with a burn-in of 500,000 followed by 4.5 million iterations. The most likely K was identified using the mean and standard deviation of Pr (X|K), and the ΔK method from Evanno, Regnaut & Goudet (2005), as calculated by the STRUCTURE HARVESTER (Earl & vonHoldt, 2011). We used CLUMPP 1.1 (Jakobsson & Rosenberg, 2007) and DISTRUCT 1.1 (Rosenberg, 2004) to align and visualize the results of the ten replicates at the most likely value of K. We also used the ‘spatial clustering of groups’ module in BAPS 5.2 to identify the best value of K using predefined sampling sites and spatial coordinates as prior information (Corander, Sirén & Arjas, 2008).

Results & Discussion

All loci were in linkage equilibrium across the entire dataset and within each population. However, all loci deviated from HWE across the entire dataset (Table 1), most likely due to a Wahlund effect resulting from population structure. Most loci within sites were in HWE except for three loci in Highbridge Park South (Table 2). Loci within each population exhibited no evidence of homozygote excess in MICROCHECKER due to microsatellite errors, with the exception of two markers in the Highbridge Park South population that were positive for null alleles (Doc03 and Ens06).

Table 1 Characteristics of five microsatellite loci genotyped in five NYC populations.

Locus	Allele size range	Na	NAb	NEc	HOd	HEe	HWEf	
Dau3a	124–268	122	8	1.86	0.246	0.462	***	
Dau11a	275–319	136	9	2.42	0.331	0.589	***	
Dau12a	273–405	110	17	5.3	0.573	0.811	***	
Doc03b	153–185	125	8	2.01	0.32	0.502	***	
Ens6c	120–188	127	10	2.19	0.142	0.542	***	
Notes.

a Number of individuals genotyped in five populations.

b Number of alleles.

c Number of effective alleles.

d Observed heterozygosity.

e Expected heterozygosity.

f *** significant deviation (P < 0.0001) from Hardy–Weinberg equilibrium for entire dataset.

Table 2 Genetic variation and bottlenecks among populations of northern dusky salamanders in NYC area.

Statistics were calculated both separately and combined for the north and south samples from Highbridge Park.

Site	Na	NAb	NEc	NPd	HOe	HEf	BNg	HWEh	
Highbridge North	32.4	2.4	1.24	1	0.143	0.151	0.016	–	
Highbridge South	29.0	5.2	1.65	11	0.288	0.293	0.031	Dau11, Doc3, Ens6	
Highbridge combined	61.4	5.8	1.44	13	0.18	0.229	0.016	Dau11, Doc3, Ens6	
Corson’s Brook Woods	18.8	5.4	3.58	5	0.567	0.651	0.89	Dau11, Ens6	
Reed’s Basket Willow	24.2	3.6	1.9	3	0.406	0.428	0.313	Ens6	
Watchung Reservation	19.6	4.0	2.15	5	0.398	0.425	0.031	Dau3	
Notes.

a Average number of individuals genotyped at five loci.

b Average number of alleles.

c Average number of effective alleles.

d Number of private alleles.

e Observed heterozygosity.

f Expected heterozygosity.

g P-value from bottleneck analysis.

h Loci deviating significantly (P < 0.05) from Hardy–Weinberg equilibrium.

Heterozygosity was moderately low for the NJ and Staten Island sites (HO = 0.40–0.57), but considerably lower for both north (HO = 0.14) and south (HO = 0.29) Highbridge Park in Manhattan (Table 2). We found evidence of genetic bottlenecks for both Highbridge sites and the Watchung Reservation, but not the Staten Island populations. These populations all exhibited substantially lower heterozygosity at the same loci as D. auriculatus (HO = 0.63–0.94; Croshaw & Glenn, 2003) and D. ocoee (HO = 0.95; Adams, Jones & Arnold, 2005) sampled from non-urban streams. Five microsatellite markers are predicted to have sufficient power for detecting only moderate to severe bottlenecks, but the variability of tetranucleotide microsatellites and relatively large number of individuals sampled from each population somewhat offset concerns over the number of loci (Cornuet & Luikart, 1996). Substantial population bottlenecks are plausible demographic scenarios for the Manhattan populations given that they inhabit two tiny seeps in a degraded urban secondary forest, and dusky salamanders do not occur elsewhere in Manhattan or even in neighboring counties on different landmasses. The Watchung Reservation population occurs in a relatively large, contiguous protected area, but D. fuscus may be confined to a single stream there and no known extant populations exist nearby. Larger protected areas, population sizes, and the potential ability to disperse through stream networks may have prevented substantial demographic decline in Staten Island populations.

Table 3 Pairwise FST calculated between five NYC populations (below diagonal). Values above diagonal are P-values calculated from 1,000 permutations of the data in GenAlex.

HPN = Highbridge Park North; HPS = Highbridge South; CBW = Corson’s Brook Woods; RB = Reed’s Basket Willow; WR = Watchung Reservation.

	HPN	HPS	CBW	RB	WR	
HPN	–	0.001	0.001	0.001	0.001	
HPS	0.079	–	0.001	0.001	0.001	
CBW	0.388	0.275	–	0.001	0.001	
RB	0.514	0.409	0.242	–	0.001	
WR	0.490	0.386	0.213	0.218	–	

Highbridge South had more than twice the number of private alleles compared to the other sites, and double the heterozygosity of Highbridge North (Table 2). Coupled with three loci out of Hardy–Weinberg, it is possible that Highbridge South receives migrants or contains ancestral variation from unknown seeps in the park. The possibility of human translocations also cannot be completely ruled out in a dense urban environment where these salamanders are well-known to the local human population. The MICROCHECKER results indicated a low but detectable frequency of null alleles at this site, suggesting the occurrence of mutations in the flanking sequence for two of the microsatellite loci. Highbridge North exhibits the lowest genetic diversity in NYC with most alleles nearing fixation, and may have experienced a much more severe population bottleneck than other populations. Long-term mark-recapture studies are needed to determine whether these populations are declining due to inbreeding depression. Occasional population surveys conducted over the last seven years indicate that both of these sites harbor dozens to hundreds of individuals with a broad range of body sizes (E. Pehek, unpublished data). Given no change in habitat quality, these populations likely do not face imminent extinction due to demographic factors. However, lack of connectivity with any other populations has likely resulted in strong genetic drift and loss of variation over the last several dozens of generations.

Pairwise FST was significant for all population pairs, and ranged from 0.08 between the two Highbridge sites to 0.51 between Highbridge North and Reed’s Basket Willow (Table 3). All pairwise values were greater than 0.2 except for FST between the two Highbridge sites. The Highbridge sites exhibited greater genetic differentiation from the three Staten Island/NJ sites than any of the Staten Island/NJ sites from each other. These results suggest that Highbridge Park contains the most isolated populations of dusky salamanders in the NYC region. Clustering analysis in STRUCTURE indicated that all but the two Highbridge sites contain genetically differentiated populations of dusky salamanders. The highest probability of the data, Pr (X|K), was calculated for K = 6, and the value of ΔK was highest at K = 2. The bar plot for K = 6 indicates substantial admixture between the two Highbridge sites but unique evolutionary clusters present in the three other Staten Island/NJ sites (Fig. 2). The K = 2 bar plot captures the divergence between Manhattan and the Staten Island/NJ sites, which likely predates urbanization due to the presence of these populations on different landmasses.

Figure 2 Results of evolutionary clustering analyses.

(A) bar plots from STRUCTURE analysis for estimated number of clusters K = 2 (top) and K = 6 (bottom). Sample sizes, N, appear on top of each sampling site. (B) Results of spatial clustering of groups in BAPS for K = 5. The X and Y axes represent geographic coordinates of the sampling sites. WR = Watchung Reservation (purple), CPW = Corson’s Brook Woods (blue), RB = Reed’s Basket Willow (yellow), HPS = South Highbridge Park (green), HPN = North Highbridge Park (red).

Individuals with relatively unique genotypes were also present in Highbridge South, most likely due to the high number of private alleles. Given the heterozygosity results above, D. fuscus in North Highbridge contain only a small subset of the genetic variation found south of the two bridges dividing the park. The BAPS analysis identified all five sampling sites as unique evolutionary clusters (posterior probability = 0.999; Fig. 2). This latter result indicates that the north and south populations of Highbridge Park are moderately differentiated, potentially due to the bridges separating them.

Our results indicate that isolated urban populations of stream salamanders in NYC have become genetically differentiated and contain little genetic diversity. Future work will be needed to determine whether these patterns occurred due to urbanization of NYC over the last 200 years, or predate the formation of the city. Extremely low variability for a pair of Manhattan sites separated by two bridges likely reflects an earlier or more severe isolation than for the other sites. A moderate FST between the two Manhattan sites, and several private alleles in the area south of two bridges that bisect Highbridge Park, indicate that urban infrastructure can create near-total isolation between proximal stream salamander populations. Heterozygosity was not as low in Staten Island, perhaps because these populations have not experienced recent bottlenecks and reside in a much larger system of protected areas with higher-order streams. Improvements in water quality, removal of within-stream and overland barriers could potentially increase dusky salamander habitat in Staten Island, although few opportunities would exist for such actions in Manhattan. Restoration of connectivity in Staten Island could also take the form of culverts under roads, “daylighting” of streams that currently flow underground due to development, and removal of stream impoundments. Reintroductions from larger populations distant from the urban core may also improve the long-term prospects for these Staten Island populations. However, disease screening (i.e. amphibian chytrid fungus) and further genetic analysis should be conducted on potential source populations before any reintroductions are attempted on Staten Island, as pairwise FST values were relatively high even between populations on the island. The recent discovery of a new species of leopard frog on Staten Island also raises the possibility of previously underappreciated phylogeographic structure or centers of endemism for amphibians in the NYC metropolitan area (Newman et al., 2012).

The only other published study of urban plethodontid genetics found much lower pairwise FST, but a similar heterozygosity, to values reported in this study for Staten Island and NJ (HO = 0.34–0.51; Noël & Lapointe, 2010). However, these authors focused on a terrestrial species, Plethodon cinereus, in Montréal that can persist in even the smallest woodlots; other genetic results indicate that this species is unaffected by all but the largest roadways (Marsh et al., 2007). P. cinereus occurs in many NYC parks, community gardens, and other semi-natural spaces, suggesting that it is less strongly affected by urban fragmentation (Pehek, 2007). The northern two-lined salamander, Eurycea bislineata, is a stream-dwelling species occupying a greater number of sites in NYC than D. fuscus, and thus may fall somewhere between P. cinereus and D. fuscus in maintaining heterozygosity in isolated urban fragments. We are currently examining E. bislineata, as well as reassessing D. fuscus, in NYC using high-density SNP (single nucleotide polymorphism) genotypes.

Relatively few studies have been conducted on amphibian population genetics, particularly on stream salamanders, but a recent meta-analysis reported that species with an IUCN status of “Least Concern” exhibited generally lower FST values than species in more threatened categories (Emel & Storfer, 2012). The authors interpreted this trend as suggestive of population extirpation in fragmented habitats as species first become threatened. Such a scenario would seem to apply to D. fuscus in urban and suburban habitats as most populations in the NYC metro area have disappeared, and the results here demonstrate substantial genetic differentiation between the extant populations.

Desmognathine salamanders have been the subject of many phylogeographic and molecular systematic analyses due to their high diversity in Appalachia (Bonett, 2002; Crespi, Rissler & Browne, 2003; Rissler & Taylor, 2003; Tilley, Eriksen & Katz, 2008; Beamer & Lamb, 2008), but there are only a few published population genetic analyses available to place the results of this study in context. Rissler, Wilbur & Taylor (2004) found little genetic structure between populations of D. monticola in different river drainages, and Croshaw & Glenn (2003) reported two- to four-fold higher heterozygosity in a D. auriculatus population compared to the results presented here. An Appalachian endemic with a highly restricted range, D. folkertsi, also exhibited very little genetic structure across river drainages, although this result may be partly explained by human transport of these salamanders between sites (Wooten et al., 2010). These results indicate that the low levels of genetic variability and substantial genetic structure between D. fuscus populations in NYC are not typical of the genus.

Gene flow in non-desmognathine stream salamanders is maintained by dispersal through stream networks in relatively undisturbed areas, although not necessarily everywhere in a species’ range (Trumbo et al., 2013). Gene flow also occurs along stream channels within the same catchments for some taxa, although there may be directional bias (Lowe et al., 2008) or significant genetic differentiation between populations in different catchments or river basins (Mullen et al., 2010). In contrast, levels of genetic divergence similar to those reported here may be natural features of some salamander species. FST values from 0.14–0.57 have been reported for high elevation taxa exhibiting philopatry to breeding ponds (Savage, Fremier & Shaffer, 2010) or river drainages (Mila et al., 2010). Genetic isolation is also characteristic of isolated, spring-associated Eurycea populations that cannot disperse through underground aquifers or streams (Lucas et al., 2008).

Urban populations of other small vertebrates with limited dispersal ability exhibit significant population structure, but not to the same degree as urban D. fuscus. White-footed mice (Peromyscus leucopus) sampled from 14 urban parks in NYC showed a high degree of genetic differentiation between parks, but the FST values were on the lower order of that estimated between the two Manhattan D. fuscus populations in Highbridge park (Munshi-South & Kharchenko, 2010). White-footed mice maintain high population densities and much higher genetic variation than D. fuscus in NYC (HO = 0.63–0.82), and vegetated urban corridors do allow for weak to moderate gene flow (Munshi-South, 2012). Urban populations of one passerine bird and three lizards exhibited significant genetic structure, but the FST values were in line with those reported for white-footed mice rather than dusky salamanders in NYC (Delaney, Riley & Fisher, 2010). Other studies of small passerine birds have also reported weak to moderate genetic structure among urban populations, but without evidence of severe loss of genetic variation (Bjorklund, Ruiz & Senar, 2010; Vangestel et al., 2011; Unfried, Hauser & Marzluff, 2013).

NYC’s dusky salamanders exhibit unusually low genetic variability and substantial genetic structure compared to desmognathines and other species in urban environments. However, this study could not determine the time since population divergence or bottlenecks to rule out the possibility that these genetic phenomena occurred before urbanization of the NYC area. Many salamanders including D. fuscus have large genomes (Wake, 2009), and more than five unlinked, genome-wide markers will be needed to estimate the timing of demographic events in urban populations with precision and accuracy. We are currently using reduced representation, next-generation sequencing approaches to generate high-density, genome-wide SNP genotypes (Davey et al., 2011; Peterson et al., 2012) for hundreds of individuals of two species of stream salamander in urban NYC and suburban/rural watersheds: D. fuscus and the northern two-lined salamander, Eurycea bislineata. We will then use landscape genomic approaches to model and compare connectivity between urban and rural streamscapes and landscapes. We will also use new statistical approaches to estimating population history from genetic data (Cornuet, Ravignie & Estoup, 2010) to examine the timing of divergence of stream salamander populations in relation to historical information on urbanization of NYC. Such approaches will also increase our ability to examine natural selection in urban salamander populations. Although lack of genetic variation and severe genetic drift may counteract the effects of selection in some isolated populations, urban environmental conditions exert potent selective pressure on salamanders (Brady, 2012) and selection itself contributes to microgeographic divergence (Richardson & Urban, in press). Selection favoring philopatry, relaxed anti-predator behavior, and larger body size are all possibilities in isolated urban seeps and streams that contain few predators or interspecific competitors, but are surrounded by hostile urban landscapes.

Amphibian responses to urbanization are generally negative (Hamer & McDonnell, 2008), and our results indicate that urban fragmentation results in substantial loss of genetic variability. The full importance of genetic variability and inbreeding for extinction risk are still unknown, but they are clearly of concern for remnant populations such as those under investigation here. Although not endangered, D. fuscus have undoubtedly declined throughout their range in eastern North America where urbanization dominates the landscape (Lannoo, 2005). Loss of genetic variability in populations isolated by human development may be an underappreciated cause and/or consequence of their continued decline.

We thank S.E. Harris, D. Jacob, B. Simmons, and S. Stanley for assistance in the field or laboratory, and B. Kajdacsi for helpful comments on the manuscript. The NJ Department of Environmental Protection – Division of Fish & Wildlife (Permit #SC2010130), the Watchung Reservation, and the NY State Department of Environmental Conservation (License to Collect & Possess Wildlife #1273) provided permission to collect samples.

Additional Information and Declarations

Competing Interests

Author Contributions

Animal Ethics

Field Study Permissions

Data Deposition

The authors declare that they have no competing interests.

Jason Munshi-South conceived and designed the experiments, performed the experiments, analyzed the data, contributed reagents/materials/analysis tools, wrote the paper.

Yana Zak performed the experiments, analyzed the data.

Ellen Pehek conceived and designed the experiments, performed the experiments, contributed reagents/materials/analysis tools, wrote the paper.

The following information was supplied relating to ethical approvals (i.e. approving body and any reference numbers):

All animal handling protocols were approved by the Natural Resources Group of the NYC Department of Parks and Recreation, and followed the recommendations of the Declining Amphibian Task Force’s “Fieldwork Code of Practice” (http://www.fws.gov/ventura/species_information/protocols_guidelines/docs/DAFTA.pdf) and the NY State Department of Environmental Conservation’s “Bio-safety Protocols for Reptile and Amphibian Sampling”.

The following information was supplied relating to ethical approvals (i.e. approving body and any reference numbers):

NJ Department of Environmental Protection – Division of Fish & Wildlife Permit #SC2010130

The Watchung Reservation, NJ

NY State Department of Environmental Conservation (License to Collect or Possess Wildlife #1273)

The following information was supplied regarding the deposition of related data:

Dryad DOI 10.5061/dryad.q1nc0

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
