# Peer review of "Conservation genetics of extremely isolated urban populations of the northern dusky salamander (Desmognathus fuscus) in New York City"

_PeerJ, doi:10.7717/peerj.64_

## Round 0.1 · original submission · Minor Revisions

Your paper has been reviewed by two experts in the field and both agree your paper is appropriate for the journal and generally a well-done study. However, both also make a number of comments on helping make the paper of interest to a more general audience by linking it to 'urban conservation' efforts in general and placing the study in such a context. I advise you to take to heart the extensive and thoughtful reviews.

·

Basic reporting

The basic reporting of the authors' research is clear and unambiguous. The literature cited is current and relevant to their study. Some comments on Figure 2: The text embedded in Fig. 2a is a bit difficult to read. I am not sure if these figures are intended to be published in color or in black and white, and if the former, the text might be easier to see. Either way, these letters should have more contrast with the background (e.g., a shadow effect) in order to read them more clearly. On Page 5, Line 107: use an n-dash instead of a hyphen between the range 1-10 (and elsewhere where there is a range of numbers). On Page 8, Line 178: Park should be capitalized. On Page 9, Line 186, change (i.e. chytrid fungus) to (i.e., amphibian chytrid fungus). Also, WR, HPS, etc. should be defined in the figure legend (they are not intuitive given the names presented in Fig. 2a).

Experimental design

The authors' experimental design lies within the scope of the journal. I think that the research question is not very clearly stated at the beginning of the paper, which would help readers understand more clearly what the authors were trying to accomplish. For example, why did they focus on the five sampling localities? There are other localities in the NYC region that likely have extant populations of this salamander, so why wasn't more extensive sampling performed throughout this region? It is interesting, but not surprising, that inbreeding depression is affecting these isolated and disjunct populations (what you would expect with limited migration and gene flow). However, what are the broader implications for their findings? The authors touch on the relevance of their results to the conservation of the n. dusky salamander, but it appears that there are few ways to amend the lack of biological corridors for this and other salamander populations isolated by urbanization. The authors mention that this study "could not determine the time since population divergence or bottlenecks to rule out the possibility that [genetic bottlenecks, etc.] occurred before urbanization of the NYC area". The authors suggest that a larger data set with more than five unlinked markers would be needed for this sort of investigation, but I would argue that if they held off to publish their findings until this is accomplished, that timing of demographic events would add a lot of impact and interest to this study. For one, it would potentially provide a way to test genetic divergence times with known population separation events (e.g., the construction of the Washington and Hamilton Bridges) or if this genetic structure occurred before development of the region.

Validity of the findings

The findings are interesting with regards to the influence of urbanization on Desmognathus fuscus in the New York City area. Despite the study's aforementioned limited data set, the methods of data collection and analysis used in this study are sound. The findings might be interesting to a larger audience, as this species occurs in one of the most populated areas of the United States and is therefore known by many lay persons.

Additional comments

This paper seems like a preliminary study leading to a more extensive one with greater genetic and specimen sampling. However, you have not mentioned if this will evolve into a more extensive study or not (if so, mention of these plans should be made in the conclusions).

Reviewer 2 ·

Basic reporting

Overall, the basic reporting of this manuscript is well executed. I offer several suggestions for improving the conveyance of results and contextualization of the study.
Generally, the paper could be improved by including a somewhat broader interpretation of the results as they relate to urbanization. As it stands, much of the discussion does a very nice job of relating these results to amphibians and a handful of other taxa. However, it would be worth additional, broader discussions on the implications of these results for the genetic consequences of urbanization. For example, what does this loss of genetic variation likely portend for the fate of these populations? Are the populations demographically stable? How much loss of variation can be tolerated and for how long?

Abstract
A line describing why the measured genetic parameters are of use for understanding population responses would give the paper broader appeal.

The number of populations can be reported in the abstract to qualify how many
“remnant” populations were surveyed.

Regarding the line ending “…estimates of high genetic variation within and lack of structure between populations of other desmognathine salamanders.”-- clarify this statement such that the reader understands whether this comparison is over similar spatial scales and landscape contexts. In other words, is this type of response rare for an urban amphibian separated across these distances?

Introduction
Overall, the introduction could use more background regarding how the genetic outcomes you investigated develop our understanding of these populations and, more broadly, conservation. E.g. What are the implications of genetic differentiation in this system. Also, some introduction into microsats as a suitable choice of markers would help this manuscript stand more on its own.

Line 25: qualify roads as potential (not definitive) barriers, the degree to which depends on the type and use of the road, as well as the amphibian in question.

Line 31: Elaborate on the nature of these declines. Is this a decline in richness? Population size? Both? And how much of a decline?

Line 39: Can you give more context to the sites that these remnant populations occupy? e.g. are the streams themselves surrounded entirely by development, or are they situated in a buffer of undeveloped land?

Line 42: What is the reason for population decline you mention here?

Line 48: perhaps start a new paragraph here. The flow of the introduction seems to jump.

Methods
What is the goal of your sampling design? Were these populations targeted to represent something specific? Or, as it sounds, do these sites represent most of the remaining populations. Whatever the case, be explicit about it and state why these sites were chosen.

Results and Discussion
It would be nice if Fig. 1 showed more information about the sites rather than so much of the surrounding landscape. Most readers will know that the region is highly urban. But it would be nice to see the slices of habitat in contrast to the development. Perhaps aerial image insets could be incorporated.

Can you report the actual heterozygosity values from the other studies you reference for better comparison?

Can you say something about population size?

For clarity, consider using either ‘D. fuscus’ or 'northern dusky salamander' throughout, not both.

Line 173: Qualify that your suggestion that bridges are the reason for differentiation is speculative.

Line 174: Qualify what is meant by rapidly, and how you have shown this differentiation meets the criteria for such pace.

Line 181: Elaborate a description of the Staten Island landscape as it is distinguished from the others.

Line 269: The closing sentence is a broad point, however I don’t think it is well supported in the text. Certainly, there are studies reporting loss of genetic diversity following landscape development, and it seems isolated populations would be a natural analogue or extension of such insights. Can you find a way to support your claim that this has been underappreciated in isolated populations?

Experimental design

The study is well designed and the level of inference that follows is appropriate.

Validity of the findings

No Comments

---

## Round 0.2 · accepted · Accept

I have read through your rebuttal to the reviews as well as through your revised draft. I am now comfortable accepting your paper for publication pending submission of your data to Dryad. Please be sure to submit your collection and genotype data to Dryad and provide the DOI reference in the manuscript.